# The reverse quantum limit and its implications for unconventional quantum oscillations in YbB$_{12}$

Christopher A. Mizzi[1], Satya K. Kushwaha[1,2,3,4], Priscila F. S. Rosa [5], W. Adam Phelan[3,4,6], David C. Arellano[6], Lucas A. Pressley[3,4,7], Tyrel M. McQueen [2,3,7], Mun K. Chan [1] & Neil Harrison [1] ✉

The quantum limit in a Fermi liquid, realized when a single Landau level is occupied in strong magnetic fields, gives rise to unconventional states, including the fractional quantum Hall effect and excitonic insulators. Stronger interactions in metals with nearly localized $f$-electron degrees of freedom increase the likelihood of these unconventional states. However, access to the quantum limit is typically impeded by the tendency of $f$-electrons to polarize in a strong magnetic field, consequently weakening the interactions. In this study, we propose that the quantum limit in such systems must be approached in reverse, starting from an insulating state at zero magnetic field. In this scenario, Landau levels fill in the reverse order compared to regular metals and are closely linked to a field-induced insulator-to-metal transition. We identify YbB$_{12}$ as a prime candidate for observing this effect and propose the presence of an excitonic insulator state near this transition.

Large magnetic fields have dramatic effects on electron motion[1]. In metallic systems, these effects are particularly pronounced when the cyclotron energy exceeds the Fermi energy and electrons are confined to their lowest Landau level. The resulting ground state, often termed the quantum limit[2], is highly degenerate and susceptible to instabilities yielding a rich variety of electronic phases including spin- and charge-density waves, fractional quantum Hall states, and excitonic insulators[3–8]. The likelihood of such novel phases tends to increase with the strength of electronic correlations, which makes it extremely desirable to explore the quantum limit in metals with strong electronic correlations. $f$-electron metals are an enticing platform for such studies; however, their large effective masses place the quantum limit beyond the reach of laboratory magnets[9–11]. Accordingly, quantum limit studies have largely focused on semimetals (e.g., graphite[12,13] and TaAs[14]) and two-dimensional systems (e.g., graphene[15–17] and GaAs

heterostructures[2]) where small effective masses and low carrier densities enable experimental access to the quantum limit.

In this paper, we propose the quantum limit in systems with strong electronic correlations is most readily reached in reverse, i.e., from an insulating ground state. This "reverse quantum limit" occurs when the Zeeman energy exceeds the cyclotron energy in an insulator and results in reverse Landau level filling. First, we motivate the idea that the first Landau level crossing may correspond to the lowest Landau level by presenting high-field measurements on the Kondo insulator YbB$_{12}$. We establish YbB$_{12}$ as an ideal system to investigate reverse Landau level filling because it has (1) large Zeeman and small cyclotron energies, (2) a small gap (which sets a field scale amenable to laboratory magnets), and (3) reproducible high-field, low-temperature behavior (including unconventional quantum oscillations[18–20]). Then, we demonstrate the angular dependence of the Landau level indices

[1]National High Magnetic Field Laboratory, Los Alamos National Laboratory, Los Alamos, NM 87545, USA. [2]Institute for Quantum Matter, William H. Miller III Department of Physics and Astronomy, The Johns Hopkins University, Baltimore, MD 21218, USA. [3]Department of Chemistry, The Johns Hopkins University, Baltimore, MD 21218, USA. [4]Platform for the Accelerated Realization, Analysis and Discovery of Interface Materials (PARADIM), Department of Chemistry, The Johns Hopkins University, Baltimore, MD 21218, USA. [5]MPA-Q, Los Alamos National Laboratory, Los Alamos, NM 87545, USA. [6]MST-16, Los Alamos National Laboratory, Los Alamos, NM 87545, USA. [7]Department of Materials Science and Engineering, The Johns Hopkins University, Baltimore, MD 21218, USA. ✉e-mail: nharrison@lanl.gov

matches that of the insulator–metal transition field. To explain this observed pinning to the transition field, we formally introduce the reverse quantum limit and show this angular dependence and the field-dependent frequency of the quantum oscillations are manifestations of reverse Landau level filling. Lastly, we show the reverse quantum limit framework captures key aspects of the insulating state quantum oscillations, which suggests the insulating and metallic state quantum oscillations in YbB$_{12}$ share a common origin.

## Results

YbB$_{12}$ possesses large changes in zero-field resistivity as a function of temperature ($\frac{\rho(0.5\,\text{K})}{\rho(300\,\text{K})} \sim 10^4$) consistent with small gaps of order meV at low temperatures (see Supplementary Note 2) arising from hybridization between conduction electrons and largely localized $f$-electrons[21]. When subjected to magnetic fields, YbB$_{12}$ exhibits a large, negative magnetoresistance and undergoes a bulk insulator–metal transition (Fig. 1a). An Arrhenius analysis (Fig. 1b) indicates the insulator–metal transition is driven by field-induced gap closure[22] (Fig. 1c). At first, the activation gap closes linearly with the field, followed by notable deviations from linear gap closure above ≈35 T. The linear gap closure is consistent with Zeeman splitting of the conduction and valence band edges corresponding to an effective $g$ factor $g^* = 1.9 \pm 0.1$ (Fig. 1c). The convention used throughout this work is $g^* = g_J m_J$. One possibility is that the high-field behavior may be related to crystal field level mixing[23]. However, the predicted crystal field scheme indicates gap closure at a lower field of ≈35 T[24,25]. It is also possible the non-monotonic behavior is evidence of an excitonic insulator, a possibility that will be explored in greater detail later. Note, the zero-field gap, transition field, and $g^*$ reported here are consistent with other measurements on high-quality YbB$_{12}$ single crystals[18,22,24,26].

Magnetoresistance oscillations are present in the insulating state of YbB$_{12}$ for $\mu_0 H \gtrsim 35$ T (Figs. 1a and 2a). The insulating state oscillations are periodic in inverse field with a dominant frequency ≈750 T. Frequencies were determined by indexing maxima and minima of the oscillations and then either finding the slope with respect to the inverse field (insulating state) or computing finite differences (metallic state); see Supplementary Note 9 for additional details. Temperature-dependent amplitudes are consistent with the Lifshitz–Kosevich (LK) form ($m^* \approx 7.6 m_e$, see Supplementary Note 10), which suggests these insulating oscillations are Shubnikov–de Haas (SdH) quantum oscillations[27]. SdH quantum oscillations in the field-induced metallic state (Fig. 2b) were observed using tunnel-diode oscillator (TDO) measurements, which are a contactless resistance method[28]. The metallic state quantum oscillations possess a field-dependent frequency (Fig. 2c) and, like the insulating state oscillations, have

temperature-dependent amplitudes described by the LK equation. Fitting with the LK equation indicates the metallic state effective mass is large (~10$m_e$), increases with field, and has little anisotropy (see Supplementary Note 10). Importantly, the quantum oscillations in Fig. 2 are in good agreement with previous reports of SdH and de Haas–van Alphen quantum oscillations in both the insulating[18,29,30] and metallic states[19,20] of YbB$_{12}$ (see Supplementary Note 10 for additional comparisons with the literature). This demonstrates quantum oscillations in high-quality YbB$_{12}$ are a robust and reproducible phenomenon.

The onset of the field-induced metallic state is anisotropic, increasing when the direction of the applied field is rotated away from [100] towards [011] ($H_{IM}^{[011]}/H_{IM}^{[100]} = 1.15$, Fig. 2d). We attribute this anisotropy to an anisotropic $g^*$, which is consistent with crystal field predictions[24]. For fields applied in the [100]–[011] plane, the angular dependence of the quantum oscillations measured at base temperature reveals that both the metallic and insulating state Landau level indices follow the angular dependence of the insulator–metal transition; i.e., the indices are pinned to the insulator–metal transition. Although there is some evolution in the fine structure of the oscillations, the dominant angular dependence matches the angular variation of the insulator–metal transition over a large angular and field range (Fig. 3). The relationship between the Landau level indices and the insulator–metal transition is most clearly demonstrated by tracking both phenomena at different angles (Fig. 3c); the angular dependence of each Landau level index and the insulator–metal transition collapses to a single curve when referenced to values at **H** ∥ [100] (Fig. 3d). Therefore, the Landau level indices are pinned to the insulator–metal transition.

To understand the surprising observation of Landau level indices that are tied to the metal–insulator transition, we contrast the behavior of Landau levels in a conventional metal with those of an insulator and, in doing so, introduce the reverse quantum limit. Electronic states in a conventional metal are quantized by a magnetic field into Landau levels[1,27]. When Zeeman splitting is considered, the Landau levels are given by

$$E^{\uparrow,\downarrow} - \mu = -\varepsilon_F + \frac{\hbar e}{m^*}B\left(\nu + \frac{1}{2}\right) \mp \frac{1}{2}g^*\mu_B B, \qquad (1)$$

where $E^{\uparrow,\downarrow}$ is the energy of the up/down ($-$,$+$) spin state referenced to the chemical potential $\mu$, $\varepsilon_F$ is the Fermi energy, $B$ is the magnetic field ($B \approx \mu_0 H$ for the applied magnetic fields used in this work[19]), $\hbar$ is Planck's constant, $e$ is electron charge, $m^*$ is effective mass, $\nu$ is the Landau level index, $g^*$ is an effective $g$-factor (for pseudospins of 1/2 that are renormalized by interactions), and $\mu_B$ is the Bohr magneton.

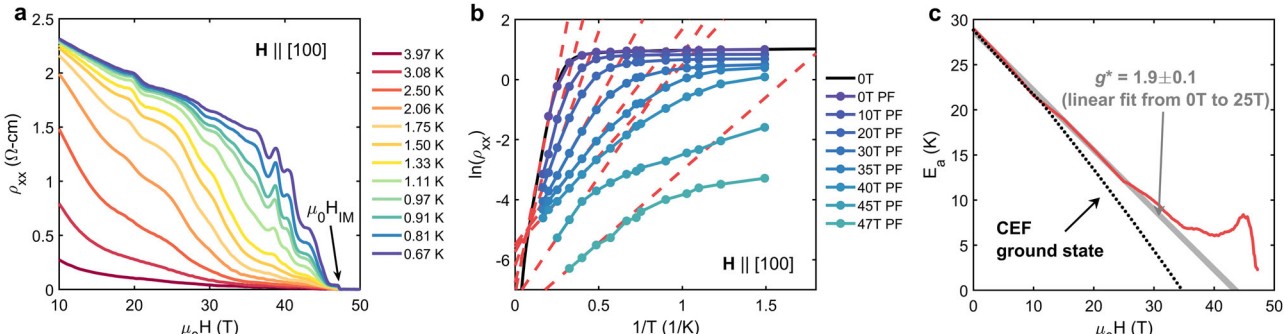

**Fig. 1 | High magnetic field behavior of YbB$_{12}$. a** YbB$_{12}$ in high magnetic fields exhibits substantial magnetoresistance, quantum oscillations, and a bulk insulator–metal transition (at $H_{IM}$). **b** Arrhenius fits (red dashed lines) to the resistivity measured in pulsed fields (PF) were used to obtain the activation gap at fixed field values. The PF data are in good agreement with zero-field resistivity measurements (black). **c** The activation gap extracted in this manner ($E_a$, red) closes

nearly linearly with a field up to 35 T, consistent with Zeeman splitting-induced gap closure (gray line). A marked departure from linear-in-field gap closure occurs above 35 T prior to the insulator–metal transition and does not appear to be consistent with crystal electric field (CEF) level mixing (black, dotted). CEF data are taken from ref. 24. See Supplementary Note 11 for additional details on gap extraction and potential sources of error.

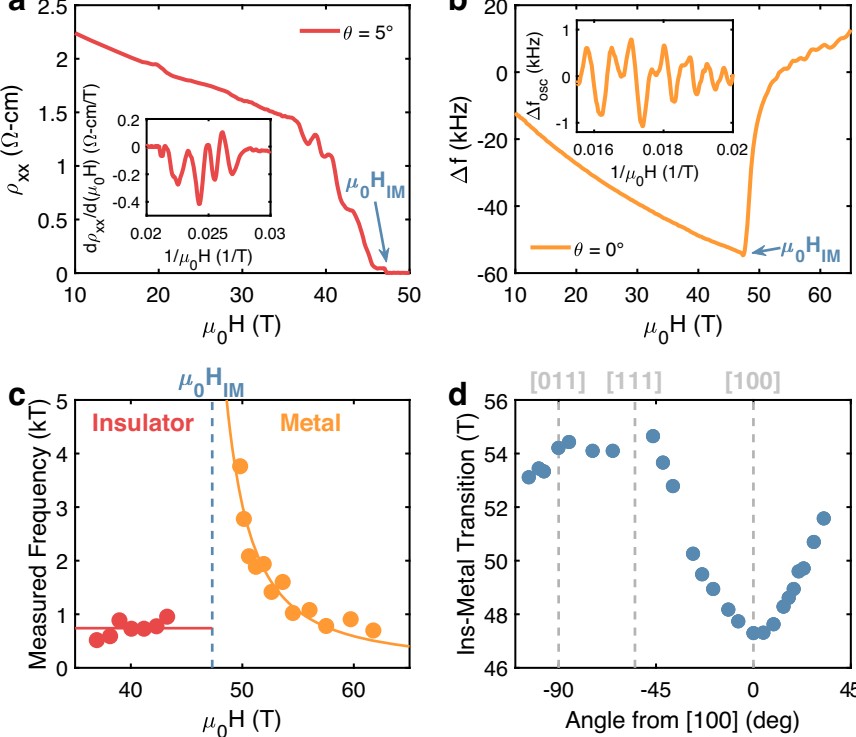

**Fig. 2 | Quantum oscillations in the insulating and field-induced metallic states of YbB₁₂. a** Magnetoresistance and **b** contactless resistance showing quantum oscillations in the insulating state and metallic state, respectively. Insets show **a** derivative of magnetoresistance and **b** background subtracted contactless resistance as functions of inverse field. **c** Insulating state quantum oscillations have a field-independent frequency of 740 ± 60 T and metallic state quantum oscillations have a field-dependent frequency (**H** ∥ [100]). Lines are guides for the eye. **d** Angular dependence of the insulator–metal transition field ($H_{IM}$). Angles correspond to rotations in the [100]–[011] plane referenced to [100]. All measurements were at ≈650 mK.

Upon increasing the field, Landau levels with decreasing indices sequentially cross the chemical potential. This process continues until the lowest Landau level is reached, corresponding to a conventional metal in the quantum limit (Fig. 4a).

Electronic states in insulators are also quantized by magnetic fields, similar to the case described by Eq. (1). For illustrative purposes, we assume a symmetric band structure with the chemical potential at the center of a zero-field gap (Δ), see Supplementary Note 3. Note, that factors such as disorder will reduce the activation gap ($E_a$, Fig. 1) from its thermodynamic value (Δ). Then, the conduction band edge ($E_C$) in the presence of a magnetic field is given by

$$E_C^{\uparrow,\downarrow} - \mu = \Delta + \frac{\hbar e}{m^*}B\left(\nu + \frac{1}{2}\right) \mp \frac{1}{2}g^*\mu_B B, \quad (2)$$

with a similar expression for the valence band edge. Landau level crossings ($E_C = \mu$) will only occur if the Zeeman energy exceeds the cyclotron energy, i.e. if $g^*\frac{m^*}{m_e} > 2$, where $m_e$ is the bare electron mass (see Supplementary Note 4). The quantity $g^*\frac{m^*}{m_e}$ can be considered a proxy for the strength of electronic correlations. When this condition is met, increasing the field causes Landau levels with increasing indices to sequentially cross the chemical potential such that the lowest Landau level is the first to cross (Fig. 4a). Therefore, the insulator is in the quantum limit at the first Landau level crossing. We call this the reverse quantum limit.

Together, $g^*\frac{m^*}{m_e}$ and Δ provides a means to classify materials according to their behavior in high magnetic fields (Fig. 4b). The quantum limit represents the most familiar case. It is realized in systems with light carriers (and/or small Zeeman energies), which have states at the chemical potential (Δ < 0). If such a system were to be gapped (Δ > 0) it would behave as a field-enhanced insulator, whereas

the introduction of stronger electronic correlations would yield a field-polarized metal. The remaining combination of $g^*\frac{m^*}{m_e}$ and Δ describes a system that is the reverse of the quantum limit, i.e., gapped at the chemical potential with heavy carriers (and/or large Zeeman energies).

Returning to YbB₁₂, we find the experimentally determined Zeeman energy ($g^* = 1.9$) and effective mass ($m^* \sim 10m_e$) satisfy the reverse quantum limit criterion ($g^*\frac{m^*}{m_e} > 2$). Furthermore, if the insulator–metal transition field, $B_{IM}$, occurs at Landau level $\nu_{IM}$, then Eq. (2) implies Landau level crossings occur when

$$\frac{1}{B_{IM}} - \frac{1}{B_\nu} = \frac{\hbar e}{\Delta m^*}(\nu - \nu_{IM}) \quad (3)$$

where $B_\nu$ is the field at which Landau level $\nu$ crosses the chemical potential. The motivation for allowing the insulator–metal transition to occur at an arbitrary Landau level is described in greater detail in Supplementary Note 5 and later in the context of a potential excitonic insulating state. Because the right-hand side of Eq. (3) only weakly depends on angle in the vicinity of the insulator–metal transition (see Supplementary Note 5), quantum oscillations are expected to be pinned to the insulator–metal transition in the reverse quantum limit, consistent with the data in Fig. 3.

Additionally, Eq. (3) corresponds to a quantum oscillation frequency,

$$|F| = \frac{m^*}{\hbar e}\Delta, \quad (4)$$

which is proportional to the product of the effective mass and hybridization gap, and analogous to the Onsager relation in a conventional metal (see Supplementary Note 5). As both of these quantities exhibit

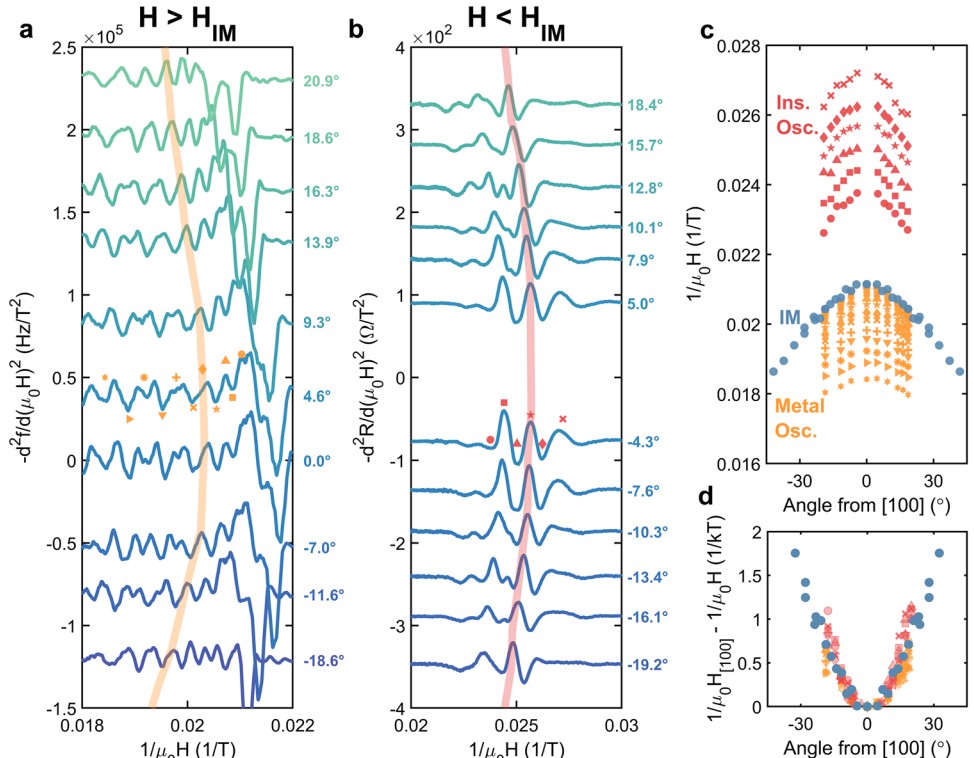

**Fig. 3 | Angular dependence of Landau level indices and the insulator–metal transition in YbB$_{12}$.** Second derivatives of **a** contactless resistance and **b** magnetoresistance as functions of the inverse field for fields applied in the [100]–[011] plane (0° corresponds to [100]). Data are vertically offset in proportion to the angle from [100]. Maxima and minima of oscillations are denoted by symbols and lines are included as guides for the eye. Second derivatives were used to avoid artifacts stemming from background subtraction, and average positions were tracked in cases of split oscillations; see Supplementary Note 9 for details. **c** Tracking the angular evolution of individual Landau level indices (different symbols) in the metallic (orange) and insulating (red) states and **d** referencing each to its value when **H** ∥ [100] demonstrates their angular dependence matches that of the insulator–metal transition (blue). All measurements were at ≈650 mK.

little angular dependence (see Supplementary Note 10), the reverse quantum limit scenario predicts a quantum oscillation frequency that does not depend on the angle. As shown in Fig. 5, the Landau level indices in the insulating state exhibit a linear relationship with an inverse field characterized by a slope that varies minimally with angle, in spite of the angular dependence of the Landau level indices, consistent with Landau levels filling in reverse. Frequencies determined by Fourier analysis corroborate this observation (Fig. 5 and Supplementary Note 5).

The field-dependent quantum oscillation frequency in the high-field metallic state (Fig. 2c) also arises naturally in the reverse quantum limit if the hybridization gap varies with the field. If Δ becomes field-dependent in the metallic state, Eq. (2) implies the quantum oscillation frequency is

$$F(B) = \frac{m^*}{\hbar e}\left(B\frac{d\Delta(B)}{dB} - \Delta(B)\right),$$ (5)

where $\Delta(B)$ denotes the field-dependent gap. Thus, the quantum oscillation frequency can exhibit a non-linear field dependence if the gap in the metallic state is a non-linear function of the field (see Supplementary Note 6). Similar considerations explain the slight deviations from the insulator-metal transition at higher angles in the metallic state in Fig. 3d (see Supplementary Note 5). Note, while Δ changes minimally in the insulating state corresponding to the zero-field gap at the chemical potential, it is also related to the spacing between the zeroth Landau levels of the conduction and valence bands for a given spin at finite field (Fig. 6a). Therefore, this quantity is well-defined in both the insulating and metallic states.

One possible origin for a non-linear Δ(B) is a situation in which the magnetic field alters the hybridization between conduction electrons and largely localized f-electrons[9,31]. A scenario in which the hybridization gap is reduced by the applied field is consistent with the appearance of non-linear magnetization in the field-induced metallic state[19,22,24] owing to partial f-electron polarization: f-electron fluctuations are suppressed by f-electron polarization at high fields, which disrupts the Kondo-type mechanism responsible for the gap[31].

To formalize the connection between the gap and extent of f-electron polarization, i.e., the non-linear portion of the magnetization in the high-field metallic state, we assume the Fermi surface area (A) is related to the non-linear component of the magnetization (M) through

$$A(B) = a\left(\frac{M(B)}{\mu_B}\right)^{2/3},$$ (6)

where a is a constant related to the degeneracy factor and the 2/3 power arises from assuming ellipsoidal pockets (see Supplementary Note 7). With this assumption, the measured quantum oscillation frequency is

$$F_m = -B^2\frac{d}{dB}\left[\frac{\hbar}{2\pi e}\frac{1}{B}a\left(\frac{M(B)}{\mu_B}\right)^{2/3}\right].$$ (7)

As shown in Fig. 6, applying Eq. (7) yields good agreement between the field dependence of our measured quantum oscillation frequencies in the metallic state and literature values for the magnetization[19] with a single tunable parameter (a ≈ 3). Fixing this a value also gives the field-dependence of the Landau indices and hybridization gap (see

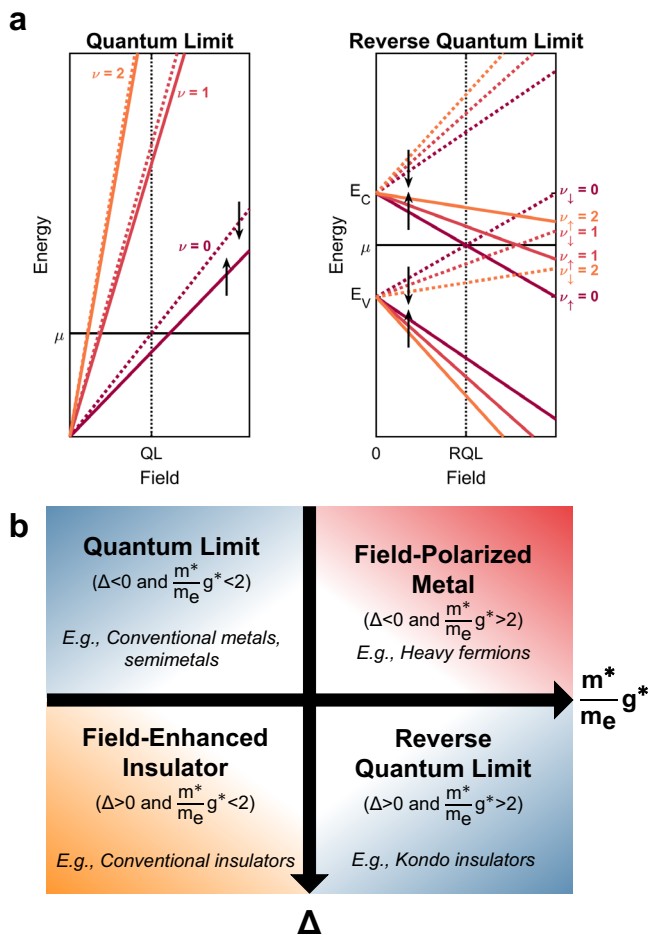

**Fig. 4 | Concept of the reverse quantum limit. a** Landau level diagram for the quantum limit (QL) and reverse quantum limit (RQL). Colors correspond to different Landau level indices and dashed/solid lines denote different spins. A conventional metal reaches the QL when the cyclotron energy exceeds the Fermi energy. An insulator with a Zeeman energy greater than the cyclotron energy is similar, except the Landau levels cross the chemical potential in reverse. **b** High-field electronic phase diagram considering if the electronic structure at the chemical potential is gapped ($\Delta > 0$) or metallic ($\Delta < 0$), and the strength of electronic correlations ($g^* \frac{m^*}{m_e}$).

Supplementary Note 7). An example of the non-linear field-dependence of the Landau levels implied by Eq. (6) is depicted in Fig. 6a.

## Discussion

The above analysis shows the reverse quantum limit explains the quantum oscillations in the high-field metallic state of YbB$_{12}$: the metallic Landau level indices track the angular dependence of the insulator–metal transition because they arise from the Landau quantization of bulk bands in an insulator which has a Zeeman energy which exceeds the cyclotron energy, and they exhibit a field-dependent frequency owing to the reduction of the hybridization gap. In other words, the Landau indexing and the insulator-to-(semi)metal transition are inextricably linked in the reverse quantum limit scenario; such a link would not exist were the carriers contributing to metallic conduction, not from Landau-quantized states.

Moreover, the insulating state oscillations exhibit a similar connection to the insulator–metal transition (Fig. 3) which strongly suggests they are a bulk phenomenon and manifestations of the same bulk Landau levels responsible for the metallic state quantum oscillations. Using Eq. (4) with the measured mass and quantum oscillation frequency in the insulating state yields $\Delta \sim 10$ meV, which is larger than

transport gaps (e.g., Fig. 1), but consistent with optical gaps[32]. Since factors such as disorder can reduce the transport gap from its thermodynamic value, our reverse quantum limit interpretation suggests the oscillations in the insulating state are related to the thermodynamic gap.

The reverse quantum limit implies entry into the insulating regime with decreasing magnetic field (and entry into the metallic regime with increasing magnetic field) meets some of the preconditions for the realization of magnetic field-induced electronic instabilities[3–8]. One possibility is a transition to an excitonic insulator at high fields prior to the insulator-metal transition in analogy with other quantum limit systems like graphite[12,13] and TaAs[14] but with much stronger electronic correlations.

In the reverse quantum limit scenario without an excitonic phase, YbB$_{12}$ would be driven to a field-induced metallic state at the $\nu = 0$ Landau level crossing. We speculate that this original insulator–metal transition would occur near 35 T because this is the field at which quantum oscillations begin in the insulating state and gap closure is predicted from crystal-field[24] calculations (Fig. 1c). However, because YbB$_{12}$ is in a low-carrier, high-correlation regime, as the number of carriers increases with magnetic field poorly screened Coulomb interactions can cause the formation of an excitonic phase[33] If this were to occur, then the insulator–metal transition would occur at a higher magnetic field corresponding to a higher value of $\nu$ than 0. As our lowest observed Landau level in the metallic state appears to be $\nu \approx 7$ (Fig. 6), this is a possibility. Once in the metallic state, the carrier density and magnetization rapidly increase corresponding to the destruction of the excitonic phase, and the quantum oscillations begin to reflect changes in hybridization and the Fermi surface shape. Additionally, an excitonic insulator phase could gap the Landau levels in the (reverse) quantum limit[4], perhaps explaining the resistivity oscillations in the insulating phase at high fields. An excitonic transition would also explain the deviations from linear-in-field gap closure for $\mu_0 H \gtrsim 35$ T (Fig. 1c) and is consistent with the insulator–metal transition being first order[24,34]. While we do not observe deviations from LK behavior (predicted for some scenarios with quantum oscillations arising in excitonic insulators[35–38]), these deviations may only be measurable at lower temperatures than were attainable in these experiments (see Supplementary Note 10). Additional experiments are needed to explore these hypotheses.

There are currently numerous proposals for the origin of the unconventional, insulating state quantum oscillations in YbB$_{12}$ which invoke concepts such as charge-neutral Fermi surfaces[39–42] and non-Hermitian in-gap states[43]. Our work indicates a common set of bulk Landau levels drive the insulating and metallic oscillations; however, exactly how these Landau levels yield resistivity oscillations while the bulk remains electrically insulating remains to be addressed. Future work will be needed to understand the consequences of the reverse quantum limit on alternative proposals.

In summary, we have introduced the reverse quantum limit, a quantum limit analogue in insulators that occurs when the Zeeman energy exceeds the cyclotron energy, and identified strongly correlated insulators as a promising platform to explore the rich array of electronic phases expected to arise in the (reverse) quantum limit. The Kondo insulator YbB$_{12}$ is an ideal system to observe this phenomenon because it possesses an electronic structure that satisfies the reverse quantum limit criterion while having an exceptionally small gap that can be closed by fields accessible in the laboratory. Predictions from the reverse quantum limit model are shown to explain how Landau level indices are pinned to the insulator-metal transition and have a field-dependent frequency in the metallic state. While a detailed mechanism for the insulating state quantum oscillations remains elusive, our work suggests they originate from the same bulk Landau levels as the metallic quantum oscillations, and provides important empirical benchmarks for theoretical descriptions of quantum

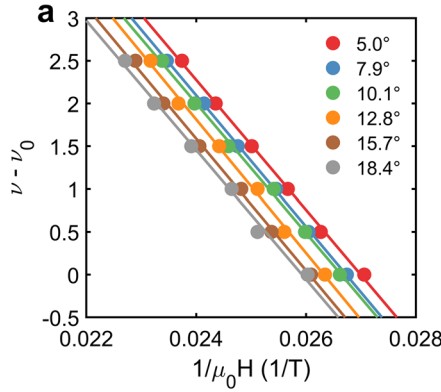
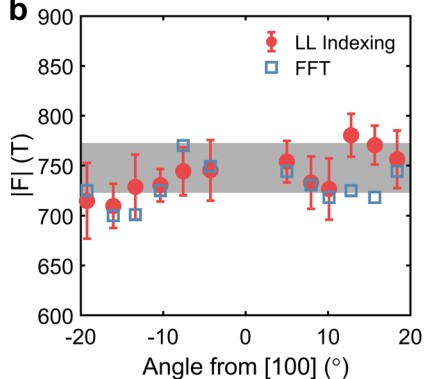

**Fig. 5 | Angular dependence of the quantum oscillation frequency in the insulating state of YbB$_{12}$. a** Landau level indices ($\nu$) as a function of the inverse field in the insulating state for the magnetic field applied along different angles with respect to [100]. Landau levels were indexed by tracking maxima and minima of the oscillations and are reported relative to the first observed oscillation. The indices follow the reverse quantum limit convention, i.e., lower Landau levels occur at lower fields. Linear fits to the data (solid lines) show nearly identical slopes. **b** Quantum oscillation frequencies from Landau level (LL) indexing and Fourier transforms (FFT) of the second derivatives of magnetoresistance (see Supplementary Note 5 for details) show minimal variation with angle. All measurements were at ≈650 mK. Error bars correspond to the 95% confidence bounds in the linear fit.

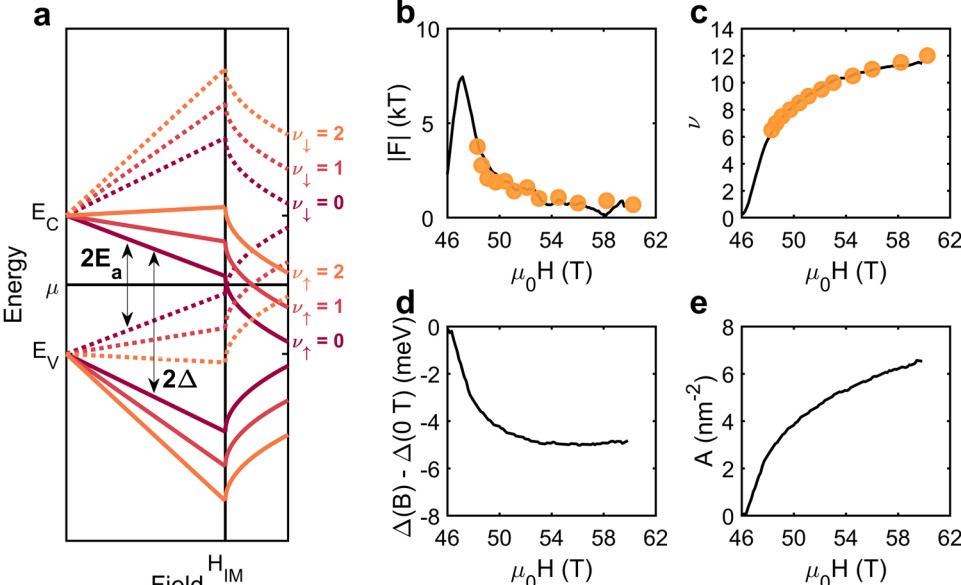

**Fig. 6 | Relating the quantum oscillation frequency to magnetization, hybridization gap, and Fermi surface area. a** Landau-level plot considering the combined effects of Zeeman splitting, Landau quantization, and gap reduction at high fields. Colors correspond to different Landau level indices and dashed/solid lines denote different spins. Above the insulator–metal transition, the Landau levels have a non-linear field dependence owing to the non-linear magnetization ($M$). **b** Quantum oscillation frequency ($F$) computed assuming the Fermi surface area is proportional to $M^{2/3}$ (black) compared to the measured quantum oscillation frequency in the metallic state (orange). From this relationship, it is possible to determine the **c** Landau indices ($\nu$) of the quantum oscillations, and the field-dependence of the **d** gap ($\Delta(B)-\Delta(0T)$) and **e** Fermi surface area ($A$). Magnetization data is taken from ref. 19.

oscillations in YbB$_{12}$ by demonstrating the central role of the insulator–metal transition.

## Methods
### Growth and structural characterization
Single crystals of YbB$_{12}$ were grown via the traveling solvent (TS) method under Ar gas atmosphere in a laser diode floating zone furnace (Crystal Systems Corp.) at the Platform for the Accelerated Realization, Analysis and Discovery of Interface Materials (PARADIM) user facility at Johns Hopkins University using a similar procedure as previously documented[26]. Subsequent crystal structure analysis employed Laue and single crystal X-ray diffraction. Single crystal domains used for the high field measurements were cut from a larger boule piece upon

being oriented using a Photonic Science Laue diffractometer. A smaller single crystal of YbB$_{12}$ than those used for Laue diffraction was mounted onto a looped fiber of a goniometer using n-grease, and the goniometer was placed on a Bruker D8 Venture X-ray diffractometer equipped with Ag K$\alpha$ ($\lambda = 0.56086$ Å) radiation. The crystallographic lattice parameters, relevant data collection information, and the corresponding refinement statistics are found in Supplementary Note 1. The cubic Laue symmetry of m-3m and the observed systematic absences led to a space group selection of Fm-3m. Structure refinement was conducted using SHELX2018[44,45], where in corresponding order, the atom positions were refined, anisotropic motion was taken into account, an extinction correction was applied, and a weighting scheme was allowed to converge. Finally, it is important to note that all

data were corrected for absorption using the multi-scan method. These diffraction methods and subsequent analysis show that the TS-grown crystal was of sufficient quality needed for this study. Additional data related to the crystal growth is available at https://doi.org/10.34863/25fk-4n21.

## Pulsed-field measurements

Electrical transport and tunnel-diode oscillator (TDO) measurements were performed in the National High Magnetic Field Laboratory Pulsed Field Facility at Los Alamos National Laboratory. Experiments utilized the 65 and 75 T duplex magnets. The sample was immersed in $^4$He liquid for experiments between 1.5 and 4 K, and $^3$He liquid for experiments below 1.5 K. Temperatures above 4 K utilized $^4$He gas.

Electrical transport was performed using a conventional four-wire, lock-in technique ($\approx$265 kHz). Contacts were made using sputtered Au pads and reinforced with silver paste. Systematic sources of error including heating artifacts are discussed in Supplementary Notes 13 and 14.

TDO[28] experiments were performed on the same sample as was used for conventional electrical transport. For this experiment, a 13-turn coil (46 gauge high-purity copper wire) was tightly wrapped around the sample. GE varnish was used to secure the coil. The TDO circuit connecting to the coil around the sample possessed a resonant frequency of $\approx$30 MHz at measurement conditions. Mixing was used to bring the measured frequency down to $\approx$2 MHz.

## Data availability

Source data are provided with this paper and other data are available from the authors upon request.

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

## Acknowledgements

This work was supported by the Department of Energy (DOE) Basic Energy Sciences (BES) project "Science of 100 Tesla." The National High Magnetic Field Laboratory is funded by National Science Foundation (NSF) Cooperative Agreements No. DMR-1157490 and No. 1164477, the State of Florida, and DOE. C.A.M. and M.K.C. were supported by the LANL LDRD Program, Project No. 20210320ER. M.K.C. acknowledges support from the NSF IR/D program while serving at the National Science Foundation. Any opinions, findings, conclusions, or recommendations expressed in this material are those of the author(s) and do not necessarily reflect the views of the National Science Foundation. S.K.K. acknowledges support of the LANL Directors Postdoctoral Funding LDRD program. This work made use of the synthesis facility of the Platform for the Accelerated Realization, Analysis, and Discovery of Interface Materials (PARADIM), which is supported by the NSF under Cooperative Agreement No. DMR-2039380. Work by the Institute for Quantum Matter, an Energy Frontier Research Center, was funded by DOE, Office of Science, BES under Award No. DE-SC0019331. The authors thank Joe D. Thompson for performing the susceptibility measurements in the Supplementary Information and Boris Maiorov for helpful discussions.

## Author contributions

C.A.M., M.K.C., and N.H. wrote the manuscript with input from all co-authors. C.A.M., S.K.K., M.K.C., and N.H. designed and performed the high-field experiments. W.A.P., L.A.P., and T.M.M. synthesized the material. P.F.S.R., W.A.P., and D.C.A. performed structural characterization. P.F.S.R. made the zero-field resistivity and heat capacity measurements. All the authors participated in the planning and discussions of experiments.

## Competing interests

The authors declare no competing interests.
