## [Peer Review File · Nature Communications]

REVIEWER COMMENTS

Reviewer #1 (Remarks to the Author):

The Kondo insulator YbB12 has recently aroused great attention. YbB12 undergoes a meta-transition at a high field. YbB12 is a particularly intriguing material given its ability to display quantum oscillations (QOs) within an insulating state at low fields, close its gap at reasonably low magnetic fields, and exhibit QOs in the high-field metallic state after the gap closure. The QOs in the insulating state are quite unique, which is still mysterious. Moreover, it has been suggested that the QOs in the high-field state exhibit unconventional field dependence, which is also highly unusual.

This paper delivers an in-depth analysis of the angular dependence of resistivity oscillations within a metallic state of YbB12 and presents an interpretation of these phenomena through the lens of the reverse quantum limit.

This paper contains a few significant points. First, the challenges associated with synthesizing high-quality YbB12 samples have impeded progress in this area, and to date, studies on QOs have been mainly reported by the Michigan group. The successful production of high-quality single crystals and independent validation of QOs would be a significant advancement for the field. Second, as the origin of the QOs in the insulating state remains elusive, there is no doubt that the interpretation of unconventional QOs in the metallic state is important.

The reverse quantum limit paradigm proposed by the authors may be important, with the potential for substantial impact. However, I still hesitate to recommend the publication of the present manuscript, because its validity warrants further investigation.

1. The authors employ Equation 2 to assert that an oscillation is observed each time the left-hand side zeros out and that the oscillation's period is reliant on m^* . Nevertheless, if this interpretation holds true, the oscillation period would be dependent on m^* even in the ordinary metal state encapsulated by Equation 1, disregarding the Fermi surface shape. This assertion contradicts the Lifshitz-Kosevich (LK) formula. Though I do not claim expertise in QOs, this discrepancy may originate from an inaccurate derivation of the oscillation frequency. Quantum oscillations in metals are detected because the density of states at the Fermi surface (or its extrema) remains constant, while the density of Landau levels intensifies with the magnetic field. Consequently, as the field strength increases, the higher Landau levels lose their occupied states, leading to the Onsager relation.

By this logic, in a reverse QO scenario, the number of occupied states and the Fermi surface (or its extrema) continue to expand as the field increases above the gap closure. Oscillations are thus discernible whenever the number of carriers surpasses that in the lowest occupied Landau level. Here, the oscillation period is dictated by the relationship between the density of states of the Fermi surface (or its extrema) and the Landau level, similar to the situation in metals. Therefore, the angular dependence of the quantum oscillation frequency does not directly support the reverse quantum oscillation scenario. I believe it is crucial for the authors to address this issue.

2. In the context of this paper, it would indeed be interesting to explore what transpires when resistance oscillations are plotted against $1/(B-B^*)$, where B^* denotes the transition field, as opposed to the traditional $1/B$. This approach, previously referred to in reference 21, may yield intriguing insights. In line with the reverse quantum limit picture, it is plausible that a scenario where the period is proportional to $1/(B-B^*)$ could be consistent, depending on the field dependence of the carrier number.

3. The authors claim that the "common set of bulk Landau levels drives the insulating and metallic oscillations." Can the authors link the present results to the dHvA oscillations observed in ref. 20?

4. It might be helpful for the readers to add a description of the residual ratio of the resistivity ($R(4.2K)/R(R.T.)$ in this case) in the main text since the observation of the QOs appears to depend on this ratio.

Reviewer #2 (Remarks to the Author):

The authors studied the Kondo insulator YbB12 under magnetic fields. They measured quantum oscillations for different field angles and found that the Landau levels are tied to the insulator-metal transition. To understand this observation, they introduced the notion of reverse quantum limit where the Zeeman energy plays a crucial role and Landau levels are filled in a reverse way compared to that in conventional (semi)metals. Besides, the authors discussed the quantum oscillation frequency and argued that it can be well explained based on the reverse quantum limit picture.

The Kondo insulator YbB12 shows interesting behaviors under magnetic fields such as the quantum oscillations in the insulating regime and the insulator-metal transition. Understanding these characteristic behaviors are highly important, since it could provide a new possible view of an insulator.

In the present manuscript, it looks that the authors carefully performed experiments and experimental results are convincing. The main point is that the field angle dependence is understood based on the newly introduced notion, the reverse quantum limit.

However, I have reservations to recommend the manuscript for publication in Nature Communications. In my understanding (mostly from the abstract and summary), the manuscript proposes the reverse quantum limit as a new state of matter or a new concept in the field of systems under magnetic fields (as shown in Fig. 4b) and argued that it is indeed realized in YbB12. The concept of the reverse quantum limit is entirely based on the model Eq. (2). Therefore, the validity of this model is a central issue in this manuscript, but there are some unclear points in the model as discussed below.

1. It looks that the explanation of the reverse quantum limit in a Kondo insulator has been oversimplified. Equation (2) is a conventional Landau levels for conduction electrons with a gap, but hybridization with a valence band or f-electrons has been neglected. (The origin of the gap may be the hybridization, but it is just implicit and the gap has been added by hand.) Such a model is not a standard model in the field of Kondo insulators where hybridization between the conduction electrons and f-electrons is essential.

One can compare Fig. 4a with the numerical calculation of a periodic Anderson model, for example Fig. 4a in Ref. 37 (Zhang et al., Phys. Rev Lett. 2016, where Zeeman effect was neglected). Note that Zeeman effects were included, for example, in Knolle and Cooper, Phys. Rev. Lett. 118, 176801 (2017) and Tada, Phys. Rev. Research 2, 023194 (2020), and it was argued that the results are robust against values of the g-factors.

The Landau level spectrum such as Fig. 4a in Ref. 37 itself has been known for a long time in the context of the electron-hole hybridization in semiconductor heterostructures. Clear figures are shown in many papers, such as Jiang et al., Phys. Rev. B 95, 045116 (2017) and references therein. I don't understand the key difference between the notion of the reverse quantum limit and these known Landau levels with electron-hole hybridization.

2. As discussed in the introduction of the manuscript, an important point of the quantum limit is that many single-particle states are degenerate in a single Landau level and such degeneracy can enhance electron correlations. If I understand correctly, such enhanced correlation effects are not expected in the reverse quantum limit, because electrons occupy Landau levels mostly in the valence band.

3. Although the comparison between the experimental data and (oversimplified) model analyses may sound reasonable, I think that a similar analysis can be done with a more standard model such as a Kondo lattice or periodic Anderson model. If this is done, what will be changed and what will be unchanged in the discussion of the manuscript?

Or, if the model Eq. (2) (rather than more standard models) is expected to be highly suitable for a Kondo insulator, it should be explained in detail.

4. Additionally, the discussion (in the main text and the supplemental information) on the oscillation frequency may be reasonable to some extent, but this also depends on the oversimplified model. An improved discussion could be done with more standard models.

Reviewer #3 (Remarks to the Author):

This paper proposes a concept of the reverse quantum limit which may be induced in an insulator with a strong electronic correlation, especially when the Zeeman energy is larger than the cyclotron energy. Although the mechanism of the formation of the reverse quantum limit is rather simple, there have not been such proposals before. It is an interesting proposal and can be important to understand the unusual phenomenon of quantum oscillation in the insulating phase. I have several questions and comments that may need to be addressed before publication.

1. The following sentence, “We argue that close to the insulator-metal transition, the insulating state should be viewed through the lens of a magnetic field-induced electronic instability affecting the

lowest Landau level states.” sounds too literary. Since the excitonic insulating state is thought to play an important role, the expression may be revised using the idea of an excitonic insulator.

2. It is not clear how the Landau level index was determined to start from 6. The Landau indexes should be shown in Fig. 8 (a) at least when the angle is zero ($B//100$).

3. Why is there no data for $B//100$ in the insulating phase in Fig. 8(b)?

4. I understand that Eq.(7) in SI is only valid when $\nu=0$. Therefore, Eq.(10) is valid only when $\nu=0$, and Eqs. (11) and (12) are not correct for arbitrary ν . I guess a similar equation with Eq.(7) is OK, for any ν , but B_{IM} is different from that for $\nu=0$. We may need to define it such as B_{IM}^{ν} . I may be misunderstood. So, the authors should give clearer explanations of how we have Eq.(12) (Eq.(3) in the main text).

5. The carrier density can change with increasing magnetic fields for magnetic fields higher than the critical magnetic field. The hole and electrons are probably induced at the same time, and they may form excitons. But the number of electrons and holes should increase with increasing magnetic field. It is necessary to explain how the reverse quantum limit scenario accommodates the issue of the change in carrier density.

6. If the electronic correlation at the Landau level at a small number index is the key to making the system insulating, the original IM transition magnetic field should be lower than the one observed. Does this original transition field correspond to 35 T? If it is yes, the definition of the IM transition magnetic field needs to be more clarified.

We thank the referees for their constructive comments and positive feedback. Responses to comments are given below in **green**. We have also clarified some portions of the main manuscript and providing additional explanations in the Supplementary Information. These changes are shown in **red** in the manuscript and supplementary information files.

Reviewer #1 (Remarks to the Author):

The Kondo insulator YbB12 has recently aroused great attention. YbB12 undergoes a meta-transition at a high field. YbB12 is a particularly intriguing material given its ability to display quantum oscillations (QOs) within an insulating state at low fields, close its gap at reasonably low magnetic fields, and exhibit QOs in the high-field metallic state after the gap closure. The QOs in the insulating state are quite unique, which is still mysterious. Moreover, it has been suggested that the QOs in the high-field state exhibit unconventional field dependence, which is also highly unusual.

This paper delivers an in-depth analysis of the angular dependence of resistivity oscillations within a metallic state of YbB12 and presents an interpretation of these phenomena through the lens of the reverse quantum limit.

This paper contains a few significant points. First, the challenges associated with synthesizing high-quality YbB12 samples have impeded progress in this area, and to date, studies on QOs have been mainly reported by the Michigan group. The successful production of high-quality single crystals and independent validation of QOs would be a significant advancement for the field. Second, as the origin of the QOs in the insulating state remains elusive, there is no doubt that the interpretation of unconventional QOs in the metallic state is important.

The reverse quantum limit paradigm proposed by the authors may be important, with the potential for substantial impact. However, I still hesitate to recommend the publication of the present manuscript, because its validity warrants further investigation.

We thank the referee for acknowledging the significance of our experiments, which independently validate quantum oscillations in YbB₁₂, and the “potential for substantial impact” of the reverse quantum limit paradigm proposed in this paper. We appreciate the questions raised by the referee and address them below.

1. The authors employ Equation 2 to assert that an oscillation is observed each time the left-hand side zeros out and that the oscillation's period is reliant on m^* . Nevertheless, if this interpretation holds true, the oscillation period would be dependent on m^* even in the ordinary metal state encapsulated by Equation 1, disregarding the Fermi surface shape. This assertion contradicts the Lifshitz-Kosevich (LK) formula. Though I do not claim expertise in QOs, this discrepancy may originate from an inaccurate

derivation of the oscillation frequency. Quantum oscillations in metals are detected because the density of states at the Fermi surface (or its extrema) remains constant, while the density of Landau levels intensifies with the magnetic field. Consequently, as the field strength increases, the higher Landau levels lose their occupied states, leading to the Onsager relation.

By this logic, in a reverse QO scenario, the number of occupied states and the Fermi surface (or its extrema) continue to expand as the field increases above the gap closure. Oscillations are thus discernible whenever the number of carriers surpasses that in the lowest occupied Landau level. Here, the oscillation period is dictated by the relationship between the density of states of the Fermi surface (or its extrema) and the Landau level, similar to the situation in metals. Therefore, the angular dependence of the quantum oscillation frequency does not directly support the reverse quantum oscillation scenario. I believe it is crucial for the authors to address this issue.

Thank you for these comments. We believe that the primary source of confusion was an omission we made in writing Eq. (1) in the manuscript. We neglected to include the Fermi energy in the submitted version of the manuscript. Below we show that with this term included we recover the Onsager relation in a typical metal. Then, we address the apparent mass dependence in the case of an insulator.

First, we consider the case of a conventional metal in the presence of a magnetic field and show that our treatment recovers the Onsager relation and a quantum oscillation frequency that is mass independent. The electronic states in a conventional metal in a magnetic field are described as

$$E^{\uparrow,\downarrow} - \mu = -\varepsilon_F + \frac{\hbar e}{m^*} B \left(\nu + \frac{1}{2} \right) \mp \frac{1}{2} g^* \mu_B B,$$

Where $E^{\uparrow,\downarrow}$ is the energy of the up/down (-,+) spin state referenced to the chemical potential (μ), ε_F is the Fermi energy, B is the magnetic field, \hbar is Planck's constant, e is electron charge, m^* is effective mass, ν is the Landau level index, g^* is an effective g-factor for pseudospins of 1/2 that is renormalized by interactions), and μ_B is the Bohr magneton. Since Landau levels cross the chemical potential when $E^{\uparrow,\downarrow} - \mu = 0$, one can derive an expression for the quantum oscillation frequency ($F = \frac{dv}{d\frac{1}{B}}$) from

$$\varepsilon_F = \frac{\hbar e}{m^*} B \left(\nu + \frac{1}{2} \right) \mp \frac{1}{2} g^* \mu_B B.$$

The result of this is

$$F = \frac{m^*}{\hbar e} \varepsilon_F,$$

or, because $\varepsilon_F = \frac{\hbar^2}{2m^*} \frac{A_k}{\pi}$ where $A_k = \pi k_F^2$,

$$F = \frac{\hbar}{2\pi e} A_k$$

which is the Onsager relation. Therefore, our treatment recovers the Onsager relation, and a mass-independent frequency, in the case of conventional metals.

Next, we focus on the case of insulators. As described in Eq. (2) of the manuscript, the conduction band electronic states ($E_C^{\uparrow,\downarrow}$) in an insulator with a zero-field gap of Δ under an applied magnetic field are given by

$$E_C^{\uparrow,\downarrow} - \mu = \Delta + \frac{\hbar e}{m^*} B \left(\nu + \frac{1}{2} \right) \mp \frac{1}{2} g^* \mu_B B,$$

analogous to the above expression for a conventional metal. Performing a similar analysis gives a quantum oscillation frequency of

$$|F| = \frac{m^*}{\hbar e} \Delta.$$

However, since $\Delta \propto \frac{1}{m^*}$ (Supplementary Information Section 3) the quantum oscillation frequency is, to leading order, independent of mass in the reverse quantum limit. Hence, a mass-independent analogue to the Onsager relation is obtained, the Lifshitz-Kosevich (LK) formula applies, and the angular dependence of the quantum oscillation frequency directly supports the reverse quantum oscillation scenario.

2. In the context of this paper, it would indeed be interesting to explore what transpires when resistance oscillations are plotted against $1/(B-B^*)$, where B^* denotes the transition field, as opposed to the traditional $1/B$. This approach, previously referred to in reference 21, may yield intriguing insights. In line with the reverse quantum limit picture, it is plausible that a scenario where the period is proportional to $1/(B-B^*)$ could be consistent, depending on the field dependence of the carrier number.

Thank you for this comment. The approach used in Ref. 21 to linearize the Landau levels was to assume an "offset field" B^* leading to an expression for the quantum oscillation frequency of the form

$$F = \frac{F_0 B}{B - B^*},$$

as indicated in the above comment. In Ref. 21, the choice of this functional form is empirical, and it is speculated that the "offset field... needed to linearize the Landau diagrams bears a qualitative similarity to the gauge field in the composite fermion interpretation of the two-dimensional fractional quantum Hall effect." While this proposition is interesting, since there are so few oscillations used to linearize the data other functional forms besides the selected $1/(B - B^*)$ form could equally well describe the data. Additionally, the value of the offset field used to describe the data in Ref. 21 (41.6 T) is quite different from the transition field (47 T), making the physical significance of the offset field an open question.

3. The authors claim that the "common set of bulk Landau levels drives the insulating and metallic oscillations." Can the authors link the present results to the dHvA oscillations observed in ref. 20?

Thank you for this comment. The resistivity oscillations we observed are in good agreement with the dHvA oscillations observed in Ref. 20. Namely, Ref. 20 reported a dHvA frequency of 720T and effective mass of $\frac{m^*}{m_e} \sim 7$ when the magnetic field is close to [100]; we find a quantum oscillation frequency of 750T and $\frac{m^*}{m_e} \sim 7.6$ from the resistivity data when the magnetic field is applied along a similar crystallographic direction.

In order to emphasize the similarity between our quantum oscillation data and previous reports of both the SdH and dHvA oscillations, we have edited the manuscript to include the portion of text given below. We also point readers to the SI (Section 10) where we have included a comparison of the effective masses derived from a Lifshitz-Kosevich analysis of our quantum oscillations, and SdH and dHvA data from the literature (see Fig. SI 11).

Importantly, the quantum oscillations in Fig. 2 are in good agreement with previous reports of SdH and de Haas-van Alphen quantum oscillations in both the insulating [20, 31, 32] and metallic states [21, 22] of YbB₁₂ (see SI [19] for additional comparisons with the literature). This demonstrates quantum oscillations in high-quality YbB₁₂ are a robust and reproducible phenomenon.

4. It might be helpful for the readers to add a description of the residual ratio of the resistivity ($R(4.2K)/R(R.T.)$ in this case) in the main text since the observation of the QOs appears to depend on this ratio.

Thank you for this suggestion. We agree this is important to included and have added the residual resistivity ratio to the main text. The main text now reads:

YbB₁₂ possesses large changes in zero-field resistivity as a function of temperature ($\frac{\rho(0.5K)}{\rho(300K)} \sim 10^4$) consistent with small gaps of order meV at low temperatures (see SI) arising from hybridization between conduction electrons and largely localized *f*-electrons.

Reviewer #2 (Remarks to the Author):

The authors studied the Kondo insulator YbB₁₂ under magnetic fields. They measured quantum oscillations for different field angles and found that the Landau levels are tied to the insulator-metal transition. To understand this observation, they introduced the notion of reverse quantum limit where the Zeeman energy plays a crucial role and Landau levels are filled in a reverse way compared to that in conventional (semi)metals. Besides, the authors discussed the quantum oscillation frequency and argued that it can be well explained based on the reverse quantum limit picture.

The Kondo insulator YbB₁₂ shows interesting behaviors under magnetic fields such as the quantum oscillations in the insulating regime and the insulator-metal transition. Understanding these characteristic behaviors are highly important, since it could provide a new possible view of an insulator. In the present manuscript, it looks that the authors carefully performed experiments and experimental results are convincing. The main point is that the field angle dependence is understood based on the newly introduced notion, the reverse quantum limit.

However, I have reservations to recommend the manuscript for publication in Nature Communications. In my understanding (mostly from the abstract and summary), the manuscript proposes the reverse quantum limit as a new state of matter or a new concept in the field of systems under magnetic fields (as shown in Fig. 4b) and argued that it is indeed realized in YbB₁₂. The concept of the reverse quantum limit is entirely based on the model Eq. (2). Therefore, the validity of this model is a central issue in this manuscript, but there are some unclear points in the model as discussed below.

1. It looks that the explanation of the reverse quantum limit in a Kondo insulator has been oversimplified. Equation (2) is a conventional Landau levels for conduction electrons with a gap, but hybridization with a valence band or f-electrons has been neglected. (The origin of the gap may be the hybridization, but it is just implicit and the gap has been added by hand.) Such a model is not a standard model in the field of Kondo insulators where hybridization between the conduction electrons and f-electrons is essential.

Thank you for raising this point. Under the most general of circumstances, when a conduction band and an f-electron band hybridize in a lattice environment, the result is an indirect gap. This has been proposed on the basis of electronic structure calculations of YbB₁₂ (see for example the Figure below). We have added a new section to the SI (Section 3) to address this point.

[from Saso and Harima, arXiv:cond-mat/0302471v1]

To lowest order, the electronic dispersions in the vicinity of the minimum of the conduction band and in the vicinity of the maximum of the valence band are parabolic. When the gap is closed by the introduction of a Zeeman interaction, the result is small electron and hole pockets (see for sample panels c and d below).

[from Liu et al, J. Phys.: Condens. Matter 30

(2018) 16LT01].

Hence, one can consider the electronic dispersion undergoing Landau quantization to be two parabolic dispersions separated by a gap, as we have assumed in our Fig. 4a. Since we cannot be certain as to precisely which pocket is responsible for the quantum oscillations in YbB_{12} , as a further simplification, we consider the electron and hole effective masses to be the same. (Also, see response to subsequent questions below).

One can compare Fig. 4a with the numerical calculation of a periodic Anderson model, for example Fig. 4a in Ref. 37 (Zhang et al., Phys. Rev Lett. 2016, where Zeeman effect was neglected). Note that Zeeman effects were included, for example, in Knolle and Cooper, Phys. Rev. Lett. 118, 176801 (2017) and Tada, Phys. Rev. Research 2, 023194 (2020), and it was argued that the results are robust against values of the g-factors.

The Landau level spectrum such as Fig. 4a in Ref. 37 itself has been known for a long time in the context of the electron-hole hybridization in semiconductor heterostructures. Clear figures are shown in many papers, such as Jiang et al., Phys. Rev. B 95, 045116 (2017) and references therein. I don't understand the key difference between the notion of the reverse quantum limit and these known Landau levels with electron-hole hybridization.

Thank you for raising this point. The Landau levels with electron-hole hybridization illustrated in those manuscripts is specific to a scenario described by a Mexican hat dispersion where the conduction and f -electron bands undergoing hybridization are isotropic within a certain plane through the Brillouin zone. This has been argued to be the case at the X point in SmB_6 , and has been proposed as a possible scenario for why the reported Landau levels and quantum oscillations in SmB_6 are similar to those from

conduction band Fermi surfaces in LaB₆. It is important to note here, that as soon as the conduction and *f*-electron bands are not perfectly isotropic, the tops of the valence band and bottoms of the conduction band would occur at points in the Brillouin zone (similar to as in YbB₁₂) instead of on a ring as for the Mexican hat dispersion.

As yet, there is no evidence a hybridization between isotropic conduction and *f*-electron bands producing a Mexican hat dispersion occurs in YbB₁₂. Also, large orbits similar to those of the unhybridized conduction band Fermi surfaces of LuB₁₂ are not observed in YbB₁₂ [see Liu et al, J. Phys.: Condens. Matter 30 (2018) 16LT01 for the LuB₁₂ Fermi surface].

2. As discussed in the introduction of the manuscript, an important point of the quantum limit is that many single-particle states are degenerate in a single Landau level and such degeneracy can enhance electron correlations. If I understand correctly, such enhanced correlation effects are not expected in the reverse quantum limit, because electrons occupy Landau levels mostly in the valence band.

Thank you for raising this point. When the Zeeman interaction closes the gap, there should be equal numbers of electrons and holes as in a compensated semimetal (although the hole pockets from the valence band and the electron pockets from the conduction band will be spin polarized). Both the electron and hole pockets will be in the quantum limit at that point.

Considering the simple case of a parabolic dispersion, the *k*-space area $A_k = \pi k^2$ of the Landau level tubes at the quantum limit is given by the magnetic field. We obtain this by setting $\nu = 0$ in

$\frac{\hbar e B}{m} \left(\nu + \frac{1}{2} \right) = \frac{\hbar^2 k^2}{2m}$, from which we obtain $A_k = \frac{\pi e B}{\hbar}$. This area is the same regardless of whether the quantum limit occurs in the conventional manner or in reverse, as we propose in YbB₁₂. The main difference with the reverse quantum limit scenario we propose is that it occurs in a system where the mass *m* is already much larger, and so the quantum limit is occurring in a system that is already strongly interacting. Whether interactions are strong enough to give rise to an instability at the Fermi surface in the reverse quantum limit such as an excitonic insulator, CDW or SDW remains an open question.

3. Although the comparison between the experimental data and (oversimplified) model analyses may sound reasonable, I think that a similar analysis can be done with a more standard model such as a Kondo lattice or periodic Anderson model. If this is done, what will be changed and what will be unchanged in the discussion of the manuscript?

Or, if the model Eq. (2) (rather than more standard models) is expected to be highly suitable for a Kondo insulator, it should be explained in detail.

We thank the referee for this comment. If we use the Anderson hybridized band model, then we must introduce sufficient higher order hopping terms to produce hybridized bands mimicking those calculated for YbB₁₂ using DFT or some other electronic structure method. However, a Taylor expansion of the dispersion in the vicinity of the conduction band minimum and valence band maximum will yield bands that are parabolic to lowest order. We can demonstrate this for a simple 1D case where the conduction band is given by $\varepsilon_k = t \cos ak_x$ and the *f*-electron band is given by $\varepsilon_f = 0$. The hybridized bands are given by $\varepsilon = \frac{t}{2} \cos ak_x \pm \sqrt{\frac{t^2}{4} \cos^2 ak_x + V^2}$, where *V* is the hybridization strength. Taking the limit $V \ll |t \cos ak_x|$ at the bottom of the conduction band (where $ak'_x = \pi - ak_x = 0$) and top of the valence band (where $k_x = 0$) we obtain $E_C \approx \frac{V^2}{t} \left(1 + \frac{a^2 k_x'^2}{2} \right)$ and $E_V \approx -\frac{V^2}{t} \left(1 + \frac{a^2 k_x^2}{2} \right)$ for the conduction and valence bands,

respectively. These can be rewritten as $E_C \approx \Delta + \frac{\hbar^2 k_x'^2}{2m}$ and $E_V \approx \Delta + \frac{\hbar^2 k_x^2}{2m}$, where $\Delta \approx \frac{V^2}{t}$ and $m \approx \frac{t\hbar^2}{V^2 a^2}$.

The dependences of Δ and m on $\frac{V^2}{t}$ will be true for most general forms of ε_k in 2 or 3 dimensions.

Equation (2) in the main text is obtained by substituting $\frac{\hbar^2 k_x^2}{2m}$ with $\frac{\hbar e B}{m} \left(\nu + \frac{1}{2} \right)$ and the Zeeman term, under assumption that the conduction band minimum (and valence band maximum) occur at distinct points in k -space (i.e. not on a ring as for the Mexican hat model).

4. Additionally, the discussion (in the main text and the supplemental information) on the oscillation frequency may be reasonable to some extent, but this also depends on the oversimplified model. An improved discussion could be done with more standard models.

We thank the referee for pointing this out. The above derivation linking the parabolic band approximation to the Anderson lattice model is now included in the Supplementary Information (Section 3).

Reviewer #3 (Remarks to the Author):

This paper proposes a concept of the reverse quantum limit which may be induced in an insulator with a strong electronic correlation, especially when the Zeeman energy is larger than the cyclotron energy. Although the mechanism of the formation of the reverse quantum limit is rather simple, there have not been such proposals before. It is an interesting proposal and can be important to understand the unusual phenomenon of quantum oscillation in the insulating phase. I have several questions and comments that may need to be addressed before publication.

We thank the referee for their interest in our proposal and recognizing the importance of its implications for quantum oscillations in insulating systems. We also appreciate their questions and comments. We address them below.

1. The following sentence, “We argue that close to the insulator-metal transition, the insulating state should be viewed through the lens of a magnetic field-induced electronic instability affecting the lowest Landau level states.” sounds too literary. Since the excitonic insulating state is thought to play an important role, the expression may be revised using the idea of an excitonic insulator.

Thank you for the suggestion. We have reworded this sentence accordingly. It now reads:

“We argue that the insulating state close to the insulator-metal transition should be viewed as a magnetic field-induced electronic instability, such as an excitonic insulating state, which affects the lowest Landau levels.”

2. It is not clear how the Landau level index was determined to start from 6.

Thank you for this question. We describe how we found the Landau level indices in greater detail below. To clarify this process in the paper, we have added similar text to Supplementary Information Section 7.

Landau level indices, ν , are related to the quantum oscillation frequency, F , according to

$$F = \frac{d\nu}{d\left(\frac{1}{B}\right)}$$

or, equivalently,

$$\nu = \frac{F}{B} + \nu_0$$

where B is the magnetic field and ν_0 is a constant. We can then relate the Landau level indices to the extremal orbit area of the Fermi surface, A , using the Onsager relation ($F = \frac{\hbar}{2\pi e} A$). This yields

$$\nu = \frac{\hbar}{2\pi e} \frac{A}{B} + \nu_0.$$

Next, as motivated in the manuscript, we assume the Fermi surface area in the high-field metallic state, $A(B)$, is related to the non-linear magnetization in the high-field metallic state, $M(B)$, because $M(B)$ is a measure of the extent of f -electron polarization. More specifically we assume

$$A(B) = a \left(\frac{M(B)}{\mu_B} \right)^{2/3}$$

where a is a fit parameter related to the degeneracy factor. Using this expression for A , we can relate both the Landau level indices and quantum oscillation frequency to the non-linear magnetization. These are the two expressions given below.

$$\nu(B) = \frac{\hbar}{2\pi e} \frac{1}{B} a \left(\frac{M(B)}{\mu_B} \right)^{2/3} + \nu_0$$

$$F(B) = -B^2 \frac{d}{dB} \left[\frac{\hbar}{2\pi e} \frac{1}{B} a \left(\frac{M(B)}{\mu_B} \right)^{2/3} \right]$$

The a parameter is fit using our experimental quantum oscillation frequency in the high-field metallic state, $F(B)$, and the experimental non-linear magnetization (taken from Ref. 21). We extract $\nu(B)$ by using this a parameter and finding the value of ν_0 which gives the best agreement with the experimental non-linear magnetization (taken from Ref. 21).

The Landau indexes should be shown in Fig. 8 (a) at least when the angle is zero ($B//100$).

Owing to the configuration of the rotator used in this experiment and sample positioning, we could not access $B \parallel [100]$ during the angle-dependent magnetoresistance experiments. The closest angle we accessed was 5° from $[100]$. Fortunately, this does not substantially change the Landau indexing or our interpretation because the quantum oscillations and insulator-metal transition slowly vary with angle within $\pm 10^\circ$ from $[100]$ (see Fig. 2 and measurements in Ref. 20).

3. Why is there no data for $B//100$ in the insulating phase in Fig. 8(b)?

See previous response.

4. I understand that Eq.(7) in SI is only valid when $\nu=0$. Therefore, Eq.(10) is valid only when $\nu=0$, and Eqs. (11) and (12) are not correct for arbitrary ν . I guess a similar equation with Eq.(7) is OK, for any ν , but B_{IM} is different from that for $\nu=0$. We may need to define it such as B_{IM}^ν . I may be misunderstood. So, the authors should give clearer explanations of how we have Eq.(12) (Eq.(3) in the main text).

We thank the referee for this suggestion. In response, we have clarified our derivation in Supplementary Section S5 and generalized it. We believe that it is now more clearly demonstrated that our framework

applies to the general case that the insulator-metal transition field B_{IM} occurs at an arbitrary Landau level. For convenience, we have reproduced the relevant portion of the text from the Supplementary Information below.

In the reverse quantum limit scenario, Landau level crossings occur at magnetic fields, B_ν , which satisfy

$$2\Delta - g^* \mu_B B_\nu + \frac{2\hbar e}{m^*} B_\nu \left(\nu + \frac{1}{2} \right) = 0.$$

If the insulator-metal transition occurs at some arbitrary Landau level ν_{IM} which crosses the chemical potential at B_{IM} , then the same expression holds:

$$2\Delta - g^* \mu_B B_{IM} + \frac{2\hbar e}{m^*} B_{IM} \left(\nu_{IM} + \frac{1}{2} \right) = 0.$$

Subtracting these two equations and reducing the result yields

$$\frac{1}{B_{IM}} - \frac{1}{B_\nu} = \frac{\hbar e}{\Delta m^*} (\nu - \nu_{IM})$$

Therefore, one anticipates the quantum oscillations to be pinned to the insulator-metal transition in the reverse quantum limit.

5. The carrier density can change with increasing magnetic fields for magnetic fields higher than the critical magnetic field. The hole and electrons are probably induced at the same time, and they may form excitons. But the number of electrons and holes should increase with increasing a magnetic field. It is necessary to explain how the reverse quantum limit scenario accommodates the issue of the change in carrier density.

Thank you for this question. In the reverse quantum limit scenario without an excitonic phase, YbB_{12} is insulating and driven to a metallic state at the $\nu = 0$ Landau level. However, because YbB_{12} is in a low-carrier, high-correlation regime, as the number of carriers increases with magnetic field poorly screened Coulomb interactions can cause the formation of an excitonic phase [e.g., Halperin and Rice, *Rev. Mod. Phys.* **40**, 755 (1968)]. If this does occur, then the insulator-metal transition will occur at a higher magnetic field. Once in the metallic state, the carrier density and magnetization rapidly increase corresponding to the destruction of the excitonic phase, and the quantum oscillations begin to reflect changes in hybridization and the Fermi surface shape. We have included a discussion of the above points in the manuscript.

6. If the electronic correlation at the Landau level at a small number index is the key to making the system insulating, the original IM transition magnetic field should be lower than the one observed. Does this original transition field correspond to 35 T? If it is yes, the definition of the IM transition magnetic field needs to be more clarified.

Thank you for this question. Yes, according to our interpretation, the effects of Landau quantization effectively shift the insulator-metal transition to a higher magnetic field. We suspect that the original

transition would occur around $\sim 35\text{T}$ (for $H \parallel [100]$) because this is the field at which (1) quantum oscillations begin in the insulating state and (2) gap closure is predicted from considering crystal-field states (Fig. 1c). However, further experiments are needed to confirm this hypothesis. We have added text explaining these points to the manuscript.

REVIEWERS' COMMENTS

Reviewer #1 (Remarks to the Author):

I appreciate the authors' effort in elucidating the derivation of both the ordinary and inverse quantum oscillation frequencies in their manuscript. The topic of quantum oscillation in insulators is of great interest currently, and this paper contributes valuable insights to the field. I am inclined to recommend this paper for publication in Nature Communications, provided one concern is satisfactorily addressed.

In the revised manuscript and the response to reviewers, the authors have noted that the observed SdH frequency aligns with the dHvA frequency reported in earlier studies. While this alignment is evident and noteworthy, the discrepancy in the angular dependence of SdH compared to that in the previous study raises an important question. Understanding this difference could be important as it might offer deeper insights into the quantum oscillation phenomena in insulators. Could the authors provide a more detailed explanation or hypothesis to account for this discrepancy?

Reviewer #2 (Remarks to the Author):

The authors answered to the referees' comments in detail. Especially a derivation of the model (2) was explained in Sec. 3 in SI, and differences between the models in the previous studies and the present manuscript are discussed in the reply. The present manuscript proposes an understanding of YbB12 which is different from those in the previous studies, which is an important step for developments of magnetic field related phenomena in correlated insulators. The model (2) seems reasonable for YbB12 and, based on the careful experiments, the authors show that the reverse quantum limit described by this model is indeed realized in YbB12. The reverse quantum limit is a simple but interesting situation which has not been explored before as far as I know. The revised manuscript provides improved discussions in response to the referees. Therefore, I recommend it for publication.

Reviewer #3 (Remarks to the Author):

The proposed reverse quantum limit model is likely to explain the quantum oscillations in magnetic fields for both insulating and metallic phases. Since the model is a very simplified one, it is rather surprising that the model can reproduce the results even quantitatively with several parameters of plausible values, indicating the model may manifest the essential mechanism of quantum oscillations in YbB12.

In this paper, it is suggested that the instability of the Fermi surface at the insulator-metal transition with the quantum limit condition can result in the excitonic insulator. This is an intriguing physical problem and it will encourage some readers to study how quantum oscillations can occur there.

Although there still have been some issues to be addressed in the future, I think this work has an impact on understanding the unusual quantum oscillations found in a correlated insulator. I recommend this paper be published in Nature Communications.

We thank the referees for their positive feedback. Responses to comments are given below in **green**. In response to Reviewer #1 we have providing additional discussion in Supplementary Note 5.

Reviewer #1 (Remarks to the Author):

I appreciate the authors' effort in elucidating the derivation of both the ordinary and inverse quantum oscillation frequencies in their manuscript. The topic of quantum oscillation in insulators is of great interest currently, and this paper contributes valuable insights to the field. I am inclined to recommend this paper for publication in Nature Communications, provided one concern is satisfactorily addressed.

In the revised manuscript and the response to reviewers, the authors have noted that the observed SdH frequency aligns with the dHvA frequency reported in earlier studies. While this alignment is evident and noteworthy, the discrepancy in the angular dependence of SdH compared to that in the previous study raises an important question. Understanding this difference could be important as it might offer deeper insights into the quantum oscillation phenomena in insulators. Could the authors provide a more detailed explanation or hypothesis to account for this discrepancy?

We thank the reviewer for recommending publication and their constructive feedback throughout this process. We also appreciate the good question regarding the SdH angular dependence. In response to this question, we have added the discussion below to Supplementary Note 5.

Our SdH data in the insulating state of YbB₁₂ indicates minimal variations in the quantum oscillation frequency with angle over a 40° range. This conclusion appears to be independent of whether we compute the quantum oscillation frequency directly from Landau level indices or with a Fourier transform (Fig. 5 of the manuscript).

The only other angular dependent measurements of SdH frequencies in YbB₁₂ that we are aware of are described in Xiang *et al.*, *Science* **362**, 65–69 (2018). In that work, a pronounced angular dependence in the SdH data with a frequency of ~800T for H || [001] is reported. However, more recent work from the same group in Xiang *et al.*, *Phys. Rev. X* **12**, 021050 (2022) indicates a significantly smaller SdH frequency of ~678T for H || [001]. This newer reported frequency conflicts with the original reported value, but now agrees with the reported dHvA frequencies in both their works [*Science* **362**, 65–69 (2018) and *Phys. Rev. X* **12**, 021050 (2022)] as well our SdH frequency.

While we cannot provide a detailed explanation for the origin of this difference, we hypothesize it has to do with sample quality. The original report of the angular dependence of the SdH quantum oscillations in the insulating state of YbB₁₂ had few, small amplitude oscillations [*Science* **362**, 65–69 (2018)] compared to later reports [*Phys. Rev. X* **12**, 021050 (2022)] and our own measurements (see figure below). The number, and presence, of SdH oscillations seems to be intimately linked to sample quality. With so few oscillations (in some cases only 2 in *Science* **362**, 65–69 (2018)), it can be difficult determine a frequency with high accuracy which could lead to the observed differences. This is clearly an important hypothesis to address with future angular dependent experiments.

Reviewer #2 (Remarks to the Author):

The authors answered to the referees' comments in detail. Especially a derivation of the model (2) was explained in Sec. 3 in SI, and differences between the models in the previous studies and the present manuscript are discussed in the reply. The present manuscript proposes an understanding of YbB12 which is different from those in the previous studies, which is an important step for developments of magnetic field related phenomena in correlated insulators. The model (2) seems reasonable for YbB12 and, based on the careful experiments, the authors show that the reverse quantum limit described by this model is indeed realized in YbB12. The reverse quantum limit is a simple but interesting situation which has not been explored before as far as I know. The revised manuscript provides improved discussions in response to the referees. Therefore, I recommend it for publication.

We thank the reviewer for recommending publication and their constructive feedback throughout this process.

Reviewer #3 (Remarks to the Author):

The proposed reverse quantum limit model is likely to explain the quantum oscillations in magnetic fields for both insulating and metallic phases. Since the model is a very simplified one, it is rather surprising that the model can reproduce the results even quantitatively with several parameters of plausible values, indicating the model may manifest the essential mechanism of quantum oscillations in YbB12.

In this paper, it is suggested that the instability of the Fermi surface at the insulator-metal transition with the quantum limit condition can result in the excitonic insulator. This is an intriguing physical problem and it will encourage some readers to study how quantum oscillations can occur there.

Although there still have been some issues to be addressed in the future, I think this work has an impact on understanding the unusual quantum oscillations found in a correlated insulator. I recommend this paper be published in Nature Communications.

We thank the reviewer for recommending publication and their constructive feedback throughout this process.